# Simplified Calculation of Shear Rotations for First-Order Shear Deformation Theory in Deep Bridge Beams

**Seyyedbehrad Emadi** [1] , **Haiying Ma** [2] , **Jose Antonio Lozano-Galant** [3],* and **Jose Turmo** [1]

1    Department of Civil and Building Engineering, Polytechnic University of Catalonia (BarcelonaTech),
     08034 Barcelona, Spain
2    Department of Bridge Engineering, Tongji University, Shanghai 200092, China
3    Department of Civil Engineering, Castilla-La Mancha University, 13001 Ciudad Real, Spain
*    Correspondence: joseantonio.lozano@uclm.es

**Abstract:** Nodal rotations are produced by bending and shear effects and bending rotations can be easily calculated using Euler–Bernoulli's stiffness matrix method. Nevertheless, shear rotations are traditionally neglected, as their effects are practically negligible in most structures. This assumption might lead to significant errors in the simulation of the rotations in some structures, as well as the wrong identification of the mechanical properties in inverse analysis. Despite its important role, no other works studying the calculation of shear rotations in deep beams were found in the literature. To fill this gap, after illustrating the errors of commercial software regarding calculating the rotations in deep beams, this study proposed a simple and intuitive method to calculate shear rotations in both isostatic and statically redundant beams. The new method calculates the shear rotation for all segments separately and introduces the result to the total rotation of the structure. This method can be applied to find the shear rotation in a redundant structure as well. A parametric study was carried out to calculate slenderness ratios to determine in what structural systems the shear rotations can be neglected. In addition, the errors in the inverse analysis of deep beams were parametrically studied to determine the role of shear rotation in different structural systems. Finally, to validate the application of the method in actual structures, a construction stage of a composite bridge was analyzed.

**Keywords:** structural system identification; observability method; shear rotation; Timoshenko's beam theory; direct analysis; inverse analysis





## 1. Introduction

Structural modeling is related to the simulation of the behavior of structures. It is achieved based on processes in which physical problems are explicated into mathematical ones with the use of a series of assumptions. In the literature, several studies dealing with the modeling of the structural behavior of beams can be found [1,2]. Most of these studies are based on Euler–Bernoulli's theory [3–7]. This theory is based on the plane section deformation hypothesis, which assumes that plane sections remain in-plane and perpendicular to the beam's neutral axis after the bending deformation [8]. Therefore, shear strains producing non-planar deformations are neglected. Strictly speaking, this theory fails for most loading cases, as shear force diagrams are usually coupled with the bending moment ones. In fact, the only loading case with zero shear forces corresponds with that of a constant bending moment alongside the beam. Euler–Bernoulli's beam theory is traditionally used in slender beams, where shear deformations are significantly smaller than the flexural ones, and therefore, their effect can be neglected. Nevertheless, in structures lacking a unidimensional geometry (such as deep beams, laminated composite walls or sandwich structures), shear deformation might play an important role and its effects should be introduced into the formulation [9].

Timoshenko [10,11] was the first to deal with the shear effects in beams and his formulation was widely used in several works in the literature [12–14]. Unlike Euler–Bernoulli's beam theory, which only considers bending rotations ($w_b$), in this approach (known as Timoshenko's beam theory or first-order shear deformation theory), the total rotation is composed of two different rotations: rotation due to the bending $w_b$ and rotation due to the shear $w_s$. The rotations considered in the Euler–Bernoulli's and Timoshenko's beam theories in a support are illustrated in Figure 1a,b, respectively.

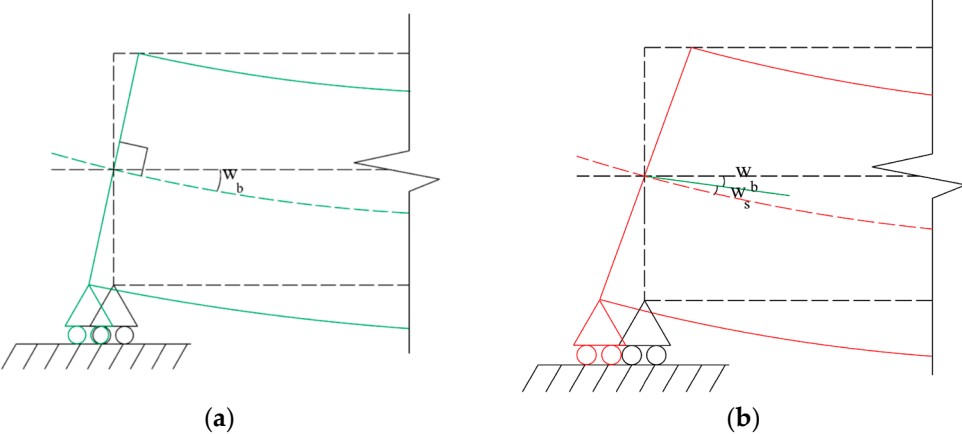

(a)          (b)

**Figure 1.** Support rotation of a beam support according to Euler–Bernoulli's (**a**) and Timoshenko's beam theories (**b**).

Timoshenko's theory was improved by Mindlin [15], who simplified the transverse shear strain as a constant distribution through the beam thickness. In this approach, a shear coefficient is used to appropriately represent the strain energy in the structures. This coefficient is used to include the effect of the non-constant shear stresses and strains in the members' cross-sections. The values of the shear coefficient for different cross-sections were proposed by Cowper [16]. In order to model the variation in shear stresses without the need for these coefficients, the finite element model (FEM) [17,18] was widely used in the literature. Thomas et al. [19] described many of the early models that deal with shear. According to these authors, the elements of these models can be classified into the following two classes: (1) simple, with two degrees of freedom at each of its two nodes, or (2) complex, with more than two degrees of freedom per node or more than two nodes per element. The first FEM for Timoshenko's beam theory was proposed by McCalley [20], who developed a two-node, four-degree-of-freedom element (transverse displacement and cross-section rotation at each node). This FEM was extended to a tapered beam by Archer [21]. Kapur [22] found that in clamped-end elements, Archer's formulation could not represent the exact boundary conditions (as the rotation due to shear was constrained to be zero). To deal with this problem, he developed a new element based on a cubic displacement function for both the bending and the shear displacements. In this formulation, the effects of shear rotations and bending rotations were analyzed separately, resulting in an eight-degree-of-freedom beam element. The FEM presented by Kapur works properly in simply supported structures (such as in simply supported beams and cantilevers). However, in more complex structures, such as statically redundant beams, Kapur's element does not work properly. Different authors used slightly different approaches for their FEMs and achieved almost the same results as Kapur. In fact, the various types of FEMs for Timoshenko's beam elements proposed in the literature [23–25] only present small differences with Kapur's formulation.

The most common variant of the FEM is the stiffness matrix method (SMM) [26,27]. This simulation procedure relates the nodal forces and the nodal displacements with the stiffness matrix of the element and it is widely used for the computer analysis of beam-like structures from both a direct [28] and inverse [29] approach. Different authors developed their own stiffness matrices to include the shear effects in the SMM formulations [30–34]. Unfortunately, these formulations only enable the analysis of the vertical deflections pro-

duced by shear and neglected the shear rotations. In order to include the effects of shear rotations in the SMM, different authors developed their own models [35–38]. In these works, specific boundary conditions were considered to remove unnecessary equations from the beam equilibrium system and specific formulations were needed. The applicability of these methods is limited to relatively simple isostatic structures under a limited number of loading cases.

Most commercial software (such as Midas/Civil [39] and many others) include Timoshenko's stiffness matrix to model the shear effects on beam-like structures. Nevertheless, as shown in Przemieniecki [34], this matrix only takes into account the effects of shear on vertical deflections. In fact, despite what could be expected, in isostatic structures, the same rotation diagrams are obtained independent of whether the shear effects are considered or not. In the case of statically redundant structures, differences in the rotation diagrams between Bernoulli and Timoshenko models are only due to the changes in the bending diagram produced by the change in reactions at the boundary conditions. These problems in the calculation of the actual rotations might also lead to significant errors in the application of inverse analysis methods to estimate the mechanical properties of the beam elements from measurements on site. This is the case with parametric methods, such as the observability technique, where Euler–Bernoulli's beam theory is traditionally considered for slender beam-like structures [40–42]. This method was improved to deal with the inverse analysis of deep beams (defined by the Eurocode EN 1992-1-1:2004 [43] as beams with less than three times the overall section depth and by the ACI committee 318 [44] as beams whose spans are equal to or less than four times the depth of the beam). To include the shear effects on the vertical deflections of these structures, the system of equations was upgraded to Timoshenko's stiffness matrix by Tomás et al. [45]. Nevertheless, this procedure presented the same limitations as the rest of the methods based on Timoshenko's stiffness matrix, as it was not able to evaluate the shear rotations. This characteristic leads to a simulation error [46] in the system of equations that jeopardizes the identification of the right structural properties. To avoid this problem, Emadi et al. [47] presented a methodology to calculate the mechanical properties of deep beams with observability techniques, avoiding the use of nodal rotations by using only the values of the vertical deflections measured on site.

Despite its important role in both the direct and inverse applications of the SMM, no other works studying the shear rotations on beams were found in the literature. To fill this gap, this paper presents the results of a detailed study of the effects of the shear rotations in both isostatic and statically redundant beams with different slenderness for direct and inverse analysis. To do so, first, a simply supported structure was analyzed to illustrate the errors of traditional structural software in the calculation of the shear rotations. Then, a new (and simple) methodology was proposed to improve the simulation of the shear rotations of computer software. In this method, the shear rotation is calculated separately and added to the bending rotation computed from the SMM. The proposed methodology was applied to determine the slenderness ratios from which the shear rotations can be neglected in different structural systems. In addition, to illustrate the important role that shear rotations play in the inverse analysis of deep beams, different structural systems were parametrically studied. Finally, to validate the application of the proposed methodology in actual structures a composite bridge was analyzed.

## 2. Materials and Methods

The nodal equilibrium equation commonly used in the stiffness matrix method [48] can be presented as

$$[K] \cdot \{\delta\} = \{f\}, \tag{1}$$

where $[K]$ is the stiffness matrix; $\{\delta\}$ is the displacements vector, which includes the horizontal, vertical and rotational displacements; and $\{f\}$ is the external force vector, which includes the horizontal forces, vertical forces and moments.

In a 2D analysis, for a six-degrees-of-freedom beam element (one horizontal deflection $u$, a vertical deflection $v$, and a rotation $w$ at the initial and final nodes) of length $L$ and constant cross-section, the common Timoshenko's stiffness matrix is shown in Equation (2):

$$[K] = \begin{bmatrix} \frac{EA}{L} & 0 & 0 & -\frac{EA}{L} & 0 & 0 \\ 0 & \frac{12EI}{L^3(1+\varnothing)} & \frac{6EI}{L^2(1+\varnothing)} & 0 & -\frac{12EI}{L^3(1+\varnothing)} & \frac{6EI}{L^2(1+\varnothing)} \\ 0 & \frac{6EI}{L^2(1+\varnothing)} & \frac{EI(4+\varnothing)}{L(1+\varnothing)} & 0 & -\frac{6EI}{L^2(1+\varnothing)} & \frac{EI(2-\varnothing)}{L(1+\varnothing)} \\ -\frac{EA}{L} & 0 & 0 & \frac{EA}{L} & 0 & 0 \\ 0 & -\frac{12EI}{L^3(1+\varnothing)} & -\frac{6EI}{L^2(1+\varnothing)} & 0 & \frac{12EI}{L^3(1+\varnothing)} & -\frac{6EI}{L^2(1+\varnothing)} \\ 0 & \frac{6EI}{L^2(1+\varnothing)} & \frac{EI(2-\varnothing)}{L(1+\varnothing)} & 0 & -\frac{6EI}{L^2(1+\varnothing)} & \frac{EI(4+\varnothing)}{L(1+\varnothing)} \end{bmatrix} \tag{2}$$

This stiffness matrix $[K]$ includes information on the axial stiffness $EA$ and the flexural stiffness $EI$, with $E$, $I$, and $A$ being the Young's modulus, inertia and area, respectively. A coefficient $\varnothing$ (shear parameter) can be found in some elements of Equation (2). This parameter is as follows:

$$\varnothing = \frac{12EI}{GA_vL^2} \tag{3}$$

where $A_v$ is the shear area and $G$ is the shear modulus written as

$$G = \frac{E}{2(1+v)} \tag{4}$$

where the constant $v$ is Poisson's ratio.

As explained in Przemieniecki [32], Timoshenko's stiffness matrix enables the calculation of vertical deformations due to shear. Nevertheless, this matrix only considers bending rotations in the system of equations, and commercial structural programs based on this formulation systematically neglect shear rotations. In fact, despite what could be expected, in isostatic structures, the same rotation diagrams are obtained independent of whether the shear effects are considered or not. To show the inability of these programs to calculate the total value of the rotations in these types of structures, an illustrative example is presented in the following section.

### 2.1. Example 1: Simply Supported Beam with a Concentrated Load

Consider a 10 m long and 0.2 m wide simply supported beam modeled with three nodes and two beam elements depicted in Figure 2a. This structure can be considered a deep beam according to the Eurocode and the ACI criteria, and the shear effects are not negligible. This beam had a constant cross-section and the value of its Young's modulus $E$, shear area $A_v$, cross-sectional area A and inertia I of the beam are 30 GPa, 0.833 m$^2$, 1 m$^2$ and 2.083 m$^4$, respectively. Dimensions of the beam cross-section are presented in Figure 2b and its mechanical properties are listed in Table 1. The boundary conditions of the structure were horizontal and vertical displacements restricted in node 1 and vertical displacement restricted in node 3 (this is to say $u_1 = v_1 = v_3 = 0$). The beam was subjected to a concentrated vertical force at mid-span (node 2) of 100 kN ($V_2 = 100$ kN).

**Table 1.** Properties of the FEM of the simply supported beam.

| Properties (Unit) | Value |
|---|---|
| Area (m$^2$) | 1.000 |
| Shear area (m$^2$) | 0.833 |
| Inertia (m$^4$) | 2.083 |
| Young's modulus (GPa) | 30.000 |
| Poisson's ratio $v$ | 0.250 |

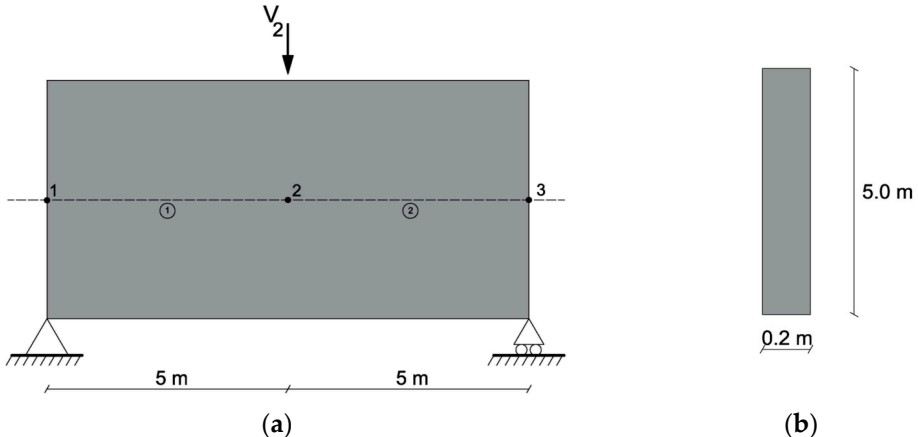

**Figure 2.** Example 1: simply supported structure: (**a**) finite element model and (**b**) cross-section.

The structure was analyzed with Midas/Civil [37] with and without considering the shear deformation effects. The obtained vertical deflections due to the bending/Bernoulli model ($v_b$) and due to the bending plus shear/Thimoshenko model ($v_b + v_s$) are summarized in Figure 3a, while the rotations for such models are presented in Figure 3b. Please note that in Figure 3b, the terms $w_{(b)}$ and $w_{(b+s)}$ stand for the rotations obtained with the Bernoulli and Timoshenko models of the structural software, respectively. Figure 3b also includes the rotations that were numerically calculated using the derivation of the vertical total deflections, including shear effects ($v_b + v_s$) throughout the beam direction (axis $x$). This information is referenced as $d(v_b + v_s)/dx$ in the figure.

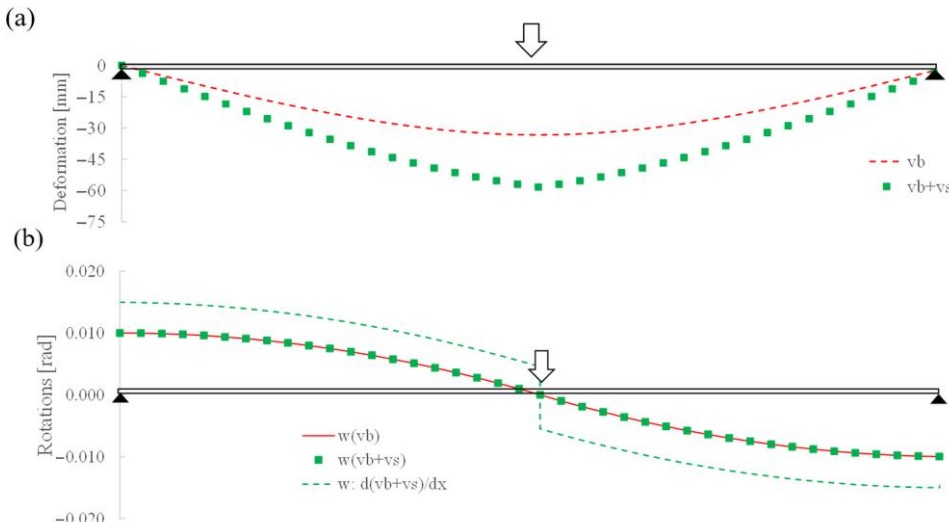

**Figure 3.** Example 1: (**a**) vertical deflections and (**b**) rotations.

As would be expected, analysis of Figure 3a shows that higher deflections were obtained when the shear deformation was considered. Nevertheless, this was not the case when the shear rotations were analyzed, as the same rotations were obtained using the commercial software with and without considering the shear deformation effects. These results are illustrated in Figure 3b, where it is shown that the rotations obtained using Midas/Civil were the same independent of whether the shear deformation effects were considered or not. The analysis of this figure shows that in this example, the commercial software failed in the calculation of the rotations, as the values obtained using the program did not correspond with those obtained using a numerical derivation of the vertical deflections. This is to highlight that these total rotations were not a continuous function, as jumps in the rotation laws can be present if point loads exist.

Timoshenko's theory can be used to calculate the deformations and rotations in the analyzed simply supported beam. According to the first order of shear deformation theory, with $S_i$ being the shear force at the $i$th node, the bending vertical deflection at mid-span (node 2) $v_{b,2}$, the vertical deflection due to shear at mid-span $v_{s,2}$, the bending rotation at the left support (node 1) $w_{b,1}$ and the shear rotation at the same support $w_{s,1}$ might be expressed using the Equations (5)–(8), respectively:

$$v_{b,2} = \frac{S_1 * L^3}{3 * E * I} \tag{5}$$

$$v_{s,2} = \frac{S_1}{2 * A_v * G} * L \tag{6}$$

$$w_{b,1} = \frac{S_1 * L^2}{4 * E * I} \tag{7}$$

$$w_{s,1} = \frac{S_1}{A_v * G} \tag{8}$$

A comparison of the vertical deflections and the rotations obtained using Timoshenko's beam theory and the results (with and without shear effects) found using Midas/Civil is summarized in Table 2.

**Table 2.** Properties of the FEM of the simply supported beam.

| Method | $v_{b,2}$ (mm) | $v_{s,2}$ (mm) | $w_{b,1}$ (rad) | $w_{s,1}$ (rad) |
|---|---|---|---|---|
| Timoshenko's beam theory | 33.34 | 20.00 | 0.010 | 0.005 |
| Midas/Civil without shear | 33.34 | 0.00 | 0.010 | 0.000 |
| Midas/Civil with shear | 33.34 | 20.00 | 0.010 | 0.000 |

Analysis of Table 2 shows that all the methods provided the same values of $v_{b,2}$ (33.34 mm) and $w_{b,1}$ (0.010 rad). When the shear deformations were considered in Midas/Civil, additional deformations $v_{s,2}$ were obtained. The values of these deformations (20.00 mm) corresponded with those calculated using Timoshenko's beam theory. Nevertheless, this was not the case for the shear rotations, as no additional rotations ($w_{s,1}$) were obtained in Midas/Civil when the shear effects were included. The error in this rotation (0.005 rad) represented the 50% bending rotation at the support ($w_{b,1}$).

The results of this example illustrate the important role that shear rotations might play and how they are systematically neglected in commercial software. In order to fill this gap, a new simulation method for the calculation of the real rotations in beams is proposed in the following section.

### 2.2. Calculation of Total Rotations with Structural Software

A new methodology is presented in this section to calculate total rotations in structures using Timoshenko's stiffness matrix method with first-order shear deformation. This methodology uses Timoshenko's beam theory to calculate the shear rotations to be added to the bending rotations calculated using the software. This is done to highlight those beam elements that converge at a given node that might have different rotations at each beam element in the same node. The bending rotations have to be the same for all the converging elements in the common node but this does not apply to the shear rotations. The proposed method is summarized as the following three steps:

- **Step 1**: calculate the bending rotations ($w_{b,i,j}$) and the shear forces ($S_{i,j}$) at each node ($i$) for each element ($j$) with the simulation software.
- **Step 2**: apply Timoshenko's beam theory to calculate the shear rotations ($w_{s,i,j}$) at each node ($i$) for each element ($j$). According to this theory, the shear rotations can be calculated from $G$, $A_v$ and $S_{i,j}$. The former two terms can be directly obtained from the

mechanical properties of the structure, while $S_{i,j}$ corresponds with the shear forces calculated in step 1.

- **Step 3**: obtain the real rotations at the $i$th node at element $j$ by summing up the bending ($w_{b,i}$) and the shear ($w_{s,i,j}$) rotations calculated in steps 1 and 2, respectively.

A summary of the procedure is shown in the flow chart in Figure 4. The shear rotations at node $i$, element $j$ are only calculated if the shear force $S_{i,j}$ is not null. Steps 2 and 3 of this flow chart can be easily implemented and programmed.

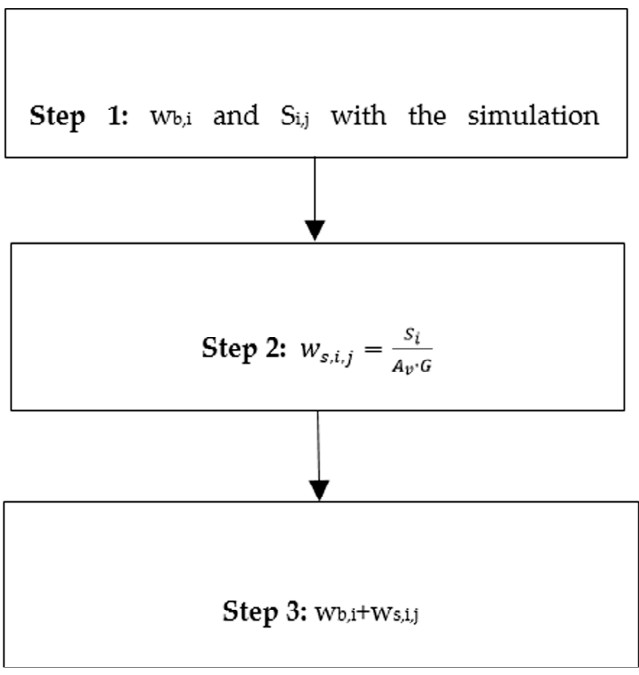

**Figure 4.** Flow chart to calculate the actual rotations in beams.

As shown in step 2, the shear rotations are proportional to the forces in the shear diagram. Hence, discontinuities in this diagram produced by concentrated loads or reactions might lead to different rotations at the same node. In addition, in accordance with Timoshenko's theory, shear rotations might appear, even at clamped supports, as they are not affected by the boundary conditions. This is a great difference from other methods [19]. Moreover, unlike other methods in the literature [35–38], the new method enables the analysis of complex structures, including redundant ones with different imposed loads. The proposed method assumes a constant value for the material's Young's modulus. Therefore, plastic rotations cannot be calculated. In future research, the method will be developed to include the effects of plastic shear rotations.

To illustrate the application of the proposed methodology in a statically redundant structure, the results of analyzing a continuous beam with a concentrated load are given below.

### 2.3. Example 2: Continuous Beam with a Concentrated Load

The analyzed beam included two 5 m long spans. The cross-section corresponded with that described in Figure 2b and the mechanical properties are described in Table 1. The model and loading case are depicted in Figure 5. The only external force applied is a concentrated vertical force at the first mid-span of 100 kN ($V$ = 100 kN).

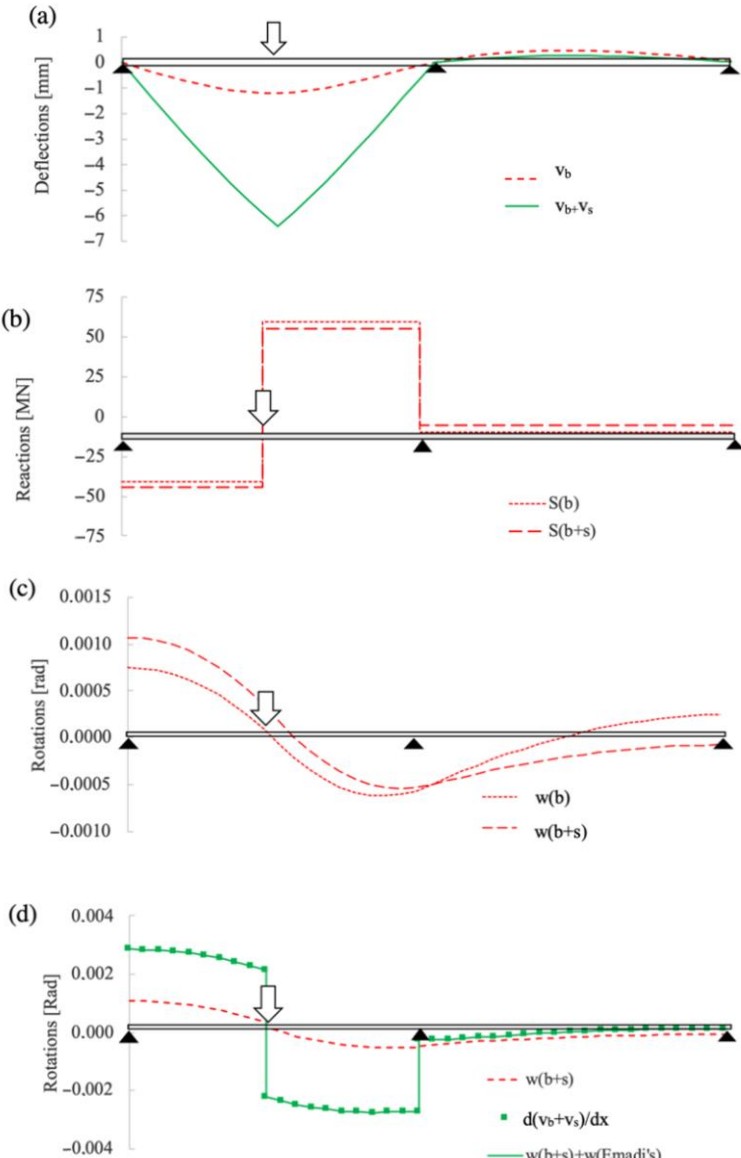

**Figure 5.** Example 2: continuous beam with and without shear effects: (**a**) vertical deflections, (**b**) shear diagrams, (**c**) rotations obtained with the commercial software, and (**d**) rotations including shear effects.

The results of the application of the proposed methodology are compared with those obtained by the computer software Midas/Civil in Figure 5. The first part of this figure (Figure 5a) includes the vertical deflections due to bending ($v_b$) and bending plus shear ($v_b + v_s$) obtained throughout the beam using Midas/Civil. Figure 5b,c include the shear diagram ($S$) and rotations ($w$) obtained using this software when bending ($S_{(b)}$ and $w_{(b)}$) or bending plus shear strains ($S_{(b+s)}$ and $w_{(b+s)}$) are considered. Finally, Figure 5d presents a comparison of the rotations calculated using Midas/Civil with those obtained using the proposed methodology when shear deformation effects were considered. This information is presented in the figure as the sum of the bending rotations obtained when the shear deformation was considered ($w_{(b+s)}$) plus the shear rotations obtained using the proposed methodology ($w$(Emadi's)). This figure also includes the rotations that were numerically calculated using the derivation of the vertical deflections, including shear effects throughout the beam directrix (axis $x$). These rotations are referenced as $d(v_b + v_s)/dx$.

The analysis of Figure 5a shows that the vertical deformations at the first mid-span (1.19 mm) increased from 1.19 mm to 6.39 mm when shear deformation was considered.

Unlike isostatic structures, in statically redundant structures, the shear deformations affect the vertical reactions at the supports. This effect is displayed in Figure 5b, where the shear diagram of the beam with and without shear deformation is presented. For example, at the first part of the beam (from left bearing to point load), the shear force S increased by 9.8% (from −40.6 MN to −44.6 MN) when shear deformation was considered. These variations on the shear forces affected the bending moment diagrams of the structure and, therefore, the bending rotations of the beam were also modified. This effect is displayed in Figure 5c. The analysis of this figure shows that the increase in the shear forces due to the introduction of the shear deformation led to higher rotations in the Midas/Civil program. For example, rotation at the left bearing was increased by 25% (from 0.0008 to 0.0010 rad) when shear effects were considered. Nevertheless, it is important to highlight that in both cases (with and without shear effects), these rotations only took into account the bending effects. Finally, Figure 5d shows how the simulation results obtained using the computer software ($w_{(b+s)}$) did not correspond with those obtained from the derivation of the beam deflections throughout the beam directrix ($x$-axis). For example, the rotations obtained using the numerical derivation at the left support (0.0029 rad) were 266.76% higher than those obtained using the computer software (0.0011 rad). Figure 5d also shows that the rotations obtained using the numerical derivation corresponded exactly with those obtained using the proposed method, where the shear rotations ($w$(Emadi's)) were added to the bending rotations calculated using the computer software.

## 3. Results

The magnitudes of the shear rotations depend, to a great extent, on the beam geometry. This section presents the results relating to a set of isostatic beams with different boundaries (simply supported or in a cantilever) and loading conditions (uniformly distributed and concentrated load) that were studied to determine the effect of the slenderness ratio on the rotations. First, the results of the direct analysis to define the error magnitude of the shear rotations and to determine the slenderness ratios from which the shear rotations can be neglected are presented. Second, the observability method [43], which is a structural inverse analysis method, was applied to illustrate how the errors in the definition of the shear rotations affected the structural system identification of the beam parameters. In this way, the effects of the shear rotations in the inverse analysis of the beams were studied. The slenderness ratios from which the shear rotations could be neglected for the correct identification of the mechanical parameters of these structures are also presented.

### 3.1. Direct Analysis

This section describes how the proposed methodology was used to calculate the shear and bending rotations of three academic examples to give a hint of the importance of shear rotation for deep beams. To do so, a simply supported beam with a concentrated load, a simply supported beam with a distributed load and a cantilever with a concentrated load were analyzed. The cross-section of all these structures corresponded with that presented in Figure 2b and the mechanical properties with those in Table 1. To evaluate the effects of the shear rotations, parametric analyses of the beam slenderness were carried out. In these analyses, the lengths of the beams ($L$) varied from 1 to 15 m while their heights ($h$) remained constant (with a value of 5 m).

The results of the simply supported beam with a concentrated load of 100 kN at mid-span ($V$ = 100 kN) are summarized in Figure 6. Figure 6a includes the bending ($w_{b,1}$) and shear ($w_{s,1}$) rotations obtained at the left support for different beam slenderness values ($L/h$). Figure 6 presents the percentage error that would result if the shear rotations were neglected. To do so, the shear rotations at node 1 were divided by the total rotations at that node ($w_{b,1} + w_{s,1}$). For illustrative purposes, the slenderness limits for deep beams proposed by the Eurocode EN [41] and by the ACI Committee 318 [42] are highlighted in this figure, together with the error limits of 2% and 5%.

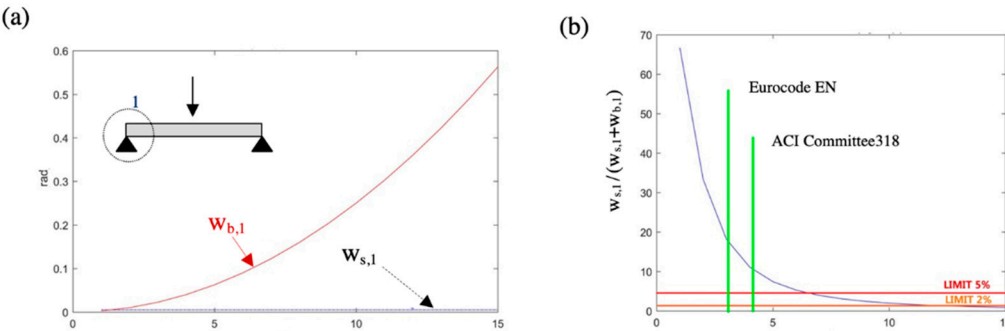

**Figure 6.** Parametric analysis of the slenderness ratio on example 1: (**a**) bending ($w_{b,1}$) and shear ($w_{s,1}$) rotations at node 1 and (**b**) percentage error of rotations in node 1 when ws,1 was neglected.

As expected, Figure 6 illustrates how the slenderer the beam, the less significant the effects of the shear rotations. On the one hand, Figure 6a shows that the shear rotation at the support ($w_{s,1}$) did not depend on the beam length, as its value (0.005 rad) remained constant for all the analyzed slenderness values. This was not the case for the bending rotations ($w_{b,1}$), as they increased nonlinearly with the length and with the slenderness ratio for a beam of constant height. In fact, these rotations varied from 0.0025 rad ($L/h$ = 1) to 0.5625 rad ($L/h$ = 15). On the other hand, Figure 6b shows that the smaller the $L/h$ ratio, the higher the error of neglecting shear rotations. In fact, shear rotations represented 68.2% of the total rotations when $L/h$ = 1. This figure also shows that a percentage error of 2% was reached for an $L/h$ ratio of 9.9. This error was increased to 5% for an $L/h$ ratio of 6.2. The geometrical definitions of deep beams in the Eurocode EN and ACI Committee 318 led to errors of 18.2% and 11.12%, respectively, when shear rotations were neglected in this example.

The results of the simply supported beam with a uniform load of 100/L kN/m applied alongside the beam are summarized in Figure 7. Figure 7a includes the bending ($w_{b,1}$) and shear ($w_{s,1}$) rotations obtained at the left support for different beam slenderness values ($L/h$). Figure 7b presents the percentage of the rotations represented by the shear rotation. This figure also includes the geometrical boundary for deep beams proposed by the Eurocode EN and by the ACI Committee 318, as well as the error limits of 2% and 5%.

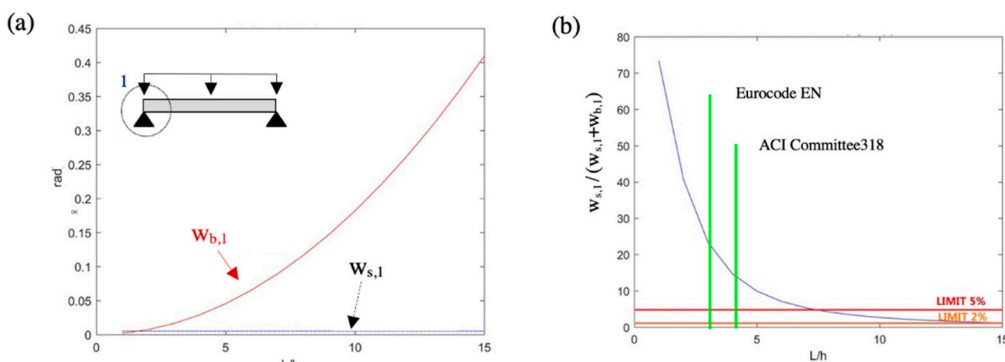

**Figure 7.** Parametric analysis of the effect of the slenderness ratio on example 2: (**a**) bending ($w_{b,1}$) and shear ($w_{s,1}$) rotations at node 1 and (**b**) percentage error of rotations in node 1 when $w_{s,1}$ was neglected.

The effects of the beam slenderness on the rotations of the simply supported beam with a uniform load were similar to those obtained for a concentrated force. In fact, as presented in Figure 7a, the shear rotation did not depend on the $L/h$ ratio, as it remained constant with a value of 0.005 rad. The bending rotations varied from 0.001 to 0.409 when the $L/h$ ratios were increased from 1 to 15. Figure 7b shows that in this example, shear rotations represented 73.1% of the rotations when $L/h$ = 1. This figure also shows that for the geometrical limitations of the deep beams presented in the Eurocode EN and

the ACI Committee 318, the errors of neglecting the shear rotations reached 23.4% and 14.7%, respectively.

The results of the analyzed cantilever beam are summarized in Figure 8. The boundary conditions of the structure referred to all the movements and rotations restricted at the left support. The loading case corresponded with a concentrated load of 100 kN applied at the free end of the beam ($V$ = 100 kN). Figure 8a,b include the same information as Figure 7a,b, respectively.

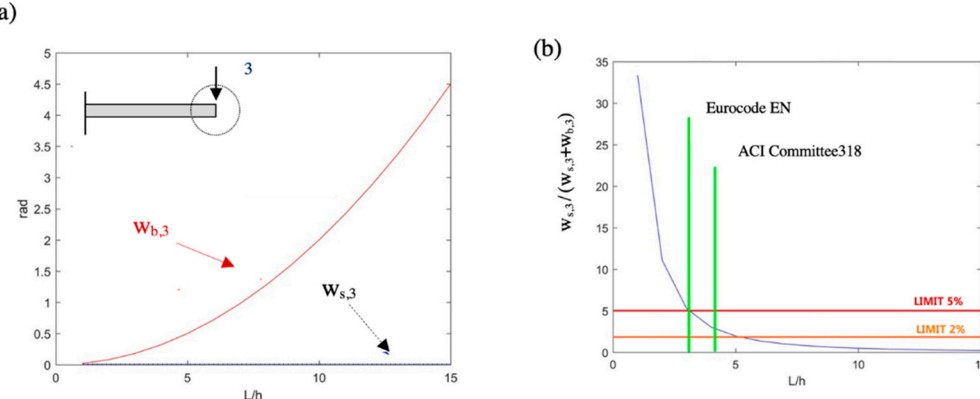

**Figure 8.** Parametric analysis of the effect of the slenderness ratio on a cantilever beam with a concentrated load: (**a**) bending ($w_{b,3}$) and shear ($w_{s,3}$) rotations at node 3 and (**b**) percentage error of rotations in node 3 when ws,3 was neglected.

Figure 8a shows that the shear rotations at the free end did not depend on the $L/h$ ratio, as they remained constant with a value of 0.010 rad. In the case of the bending rotations, they increased nonlinearly with the slenderness changing from 0.020 rad when $L/h$ = 1 to 4.501 rad when $L/h$ = 15. The analysis of Figure 8b shows that in this example, shear rotations represented 34.1% of the rotations when $L/h$ = 1. This figure also shows that for the geometrical limitations of the deep beams presented in the Eurocode EN and the ACI Committee 318, the errors of neglecting the shear rotations reached 5.3% and 3.0%, respectively. It is important to highlight that increasing the value of the imposed load increased the shear rotation value. Nevertheless, in this case, the bending rotation also increased simultaneously. For this reason, in slender beams, the value of the shear rotation remained practically negligible in comparison to the bending one. Therefore, as the beam became slender, it behaved more like an Euler–Bernoulli beam than a Timoshenko one. This same conclusion was presented by Chen and Guadalupe [49]. It is also important to highlight that, as indicated in [50], the effect of prestressing plays an important role in higher-order shear deformations. These effects on shear rotations will be addressed in future studies.

### 3.2. Inverse Analysis

This section describes how the observability method (OM) proposed by Tomás et al. [43], which is an inverse analysis method for models made out of Timoshenko beam elements, was applied to calculate the mechanical properties of academic examples to illustrate the importance of shear rotations in deep beams. The method was able to calculate the mechanical properties of beam elements from the deflections and rotations of the structure. It is important to highlight that the OM is based on Timoshenko's stiffness matrix and, therefore, the system of equations does not include the effects of the shear rotations. In fact, in this inverse analysis method, the shear deformation effects are only considered in the vertical deflections.

To evaluate the effects of the shear rotations in the inverse analysis, the three beams described in the preceding section (a simply supported beam with a concentrated load, a simply supported beam with a uniform load and a cantilever beam with a concentrated load) were calculated. It is important to highlight that introducing the shear rotations into

the OM represents a modeling error that affects the accuracy in the estimation of the beam element properties. Obviously, this error depends, to a great extent, on the magnitude of the shear rotations and, therefore, on the beam slenderness. To evaluate the effect of the slenderness of these beams, a set of parametric analyses were carried out. In these analyses, the lengths of the beams varied from 1 to 15 m while their cross-section remained constant (height, 5 m; width, 0.2 m). For the inverse analysis of each beam, two unknown parameters (the beam element inertia and the shear area) were considered. The following measurement data were considered: w1 and v2 for the simply supported beams and w3 and v3 for the cantilever beam. The different rotations introduced into the measurement data were simulated using the procedure proposed in Section 3 (as they try to simulate the rotations that would be measured on site).

Figure 9 presents the errors in the inertias estimated for the three analyzed beams for different slenderness ($L/h$) ratios. These errors were calculated as the percentage error of the estimated inertia $\bar{I}$ with respect to the actual inertia $I$ (2.083 m$^4$). For illustrative purposes, the slenderness limits for deep beams proposed by the Eurocode EN and by the ACI Committee 318 are highlighted in this figure.

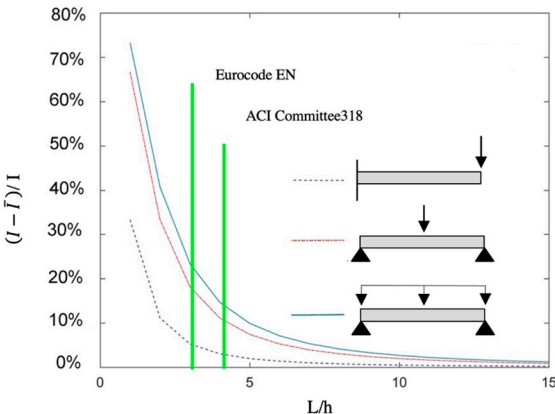

**Figure 9.** Percentage error between the estimated inertia using the OM method $\bar{I}$ with the actual one $I$ for three beams in terms of the slenderness ($L/h$ ratio).

Analysis of Figure 9 shows that shear rotations play an important role in the inverse analysis of deep beams. In fact, the lower the $L/h$ ratio, the higher the errors in the estimated inertias. For example, for the simply supported beam, the errors when $L/h = 1$ were as high as 73.2% for the uniform load and 67.9% for the concentrated one. The error reached 32.0% for the cantilever beam with the concentrated load. At the boundary of the Eurocode for deep beams, the error in the inertia estimation for the simply supported beam with the concentrated load, the simply supported beam with the distributed load, and the cantilever beam with a concentrated force were 18.2%, 23.4% and 5.3%, respectively. For the ACI geometry boundary for deep beams, the errors in these structures were reduced to 11.1%, 14.7% and 3.0%, respectively. These results illustrate that for deep beams, the effects of shear rotations should not be neglected. In contrast, in slender beams, the errors in inertia estimation were significantly lower. For example, when $L/h = 15$, the errors were below 1% in the three analyzed structures.

### 3.3. Application on the Real Bridge

To show the applicability of the proposed methodology in complex structures, an intermediate stage of the balanced cantilever construction method was analyzed and the results are presented in this section. In this construction method, the deflections of the bridge segments have to be forecasted. This makes it possible to build the segments with the proper precamber. These predeformations are updated for each construction step after a topographical survey. This data can be used by inverse analyses for model updating to increase the accuracy of the precamber calculations. According to Valerio

et al. [51], neglecting the shear effects in the balanced cantilever construction method during the erection of the first bridge segments may lead to inaccurate estimations of the bending stiffnesses.

The structure chosen for this example was a simplified model of an intermediate construction stage of the Yunbao Bridge over the Yellow River in China [52] (see Figure 10a). This bridge was erected with a novel construction technology called asynchronous pouring construction (APC), which was proposed on the basis of traditional balanced cantilever construction [53]. The Yunbao bridge has a total length of 906 m divided into the following spans: 48 m + 9 × 90 m + 48 m. The deck is continuous, and its cross-section corresponds with a composite box girder. The height of the deck ranges between 5.5 m (at the supports) and 2.7 m (at mid-span). The deck in the analyzed construction stage was 29.5 m long and it was composed of seven segments (three 4.5 m-long segments at each side joined to a central 2.5 m-long section over the pile). The formwork traveler was located at the end of each lateral span, as presented in Figure 10b. It is important to highlight that the prestressing of the balanced cantilever erection was not studied, as this load was not limited to the first-order shear deformation effects investigated in this study. To study higher-order shear deformations, the effects of the prestressing load will be studied in future works.

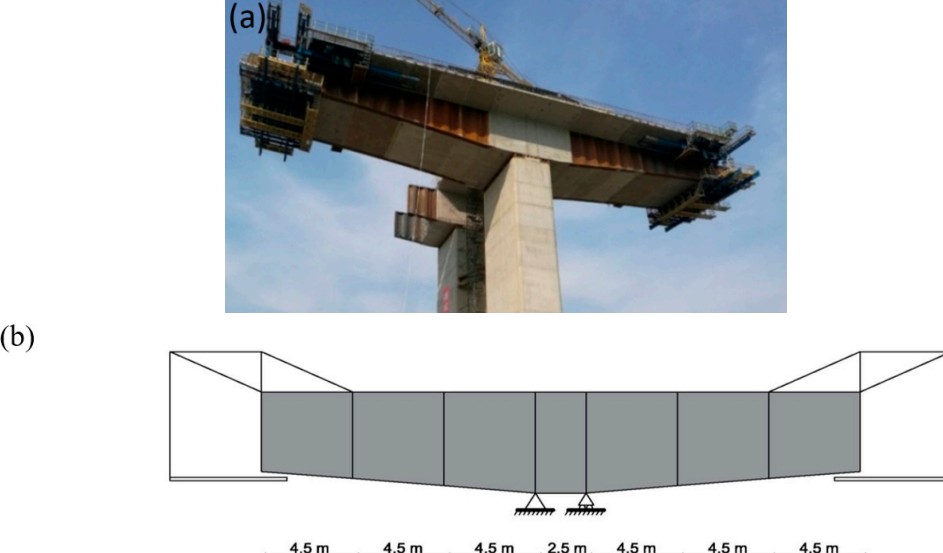

**Figure 10.** Example of application: (**a**) Yunbao Bridge under construction [52] and (**b**) a sketch of the analyzed construction stage.

A simplified geometry of the real bridge was analyzed. The connection between the concrete and the steel was assumed to be rigid, and the relative slip between both materials was neglected. The effects of prestressing on the shear rotation were overlooked in this example; these effects on shear rotations will be addressed in future studies. The cross-section of the composite deck is presented in Figure 11a and the mechanical and material properties are listed in Table 3. The FEM of this structure is presented in Figure 11b and it included eight nodes and seven beam elements. Each of these elements was defined with Young's modulus E, inertias I and shear areas $A_v$, as shown in this sub-figure.

**Table 3.** Properties of the FEM of the bridge.

| Parameters | Elements 1 and 7 | Elements 2 and 6 | Elements 3 and 5 | Element 4 |
|---|---|---|---|---|
| $EA$ (N) | $4.33 \times 10^{11}$ | $4.38 \times 10^{11}$ | $4.43 \times 10^{11}$ | $4.46 \times 10^{11}$ |
| $GA_v$ (N) | $1.10 \times 10^{10}$ | $1.25 \times 10^{11}$ | $1.43 \times 10^{11}$ | $1.53 \times 10^{11}$ |
| $EI$ (N·m$^2$) | $1.34 \times 10^{12}$ | $1.69 \times 10^{12}$ | $2.13 \times 10^{12}$ | $2.38 \times 10^{12}$ |

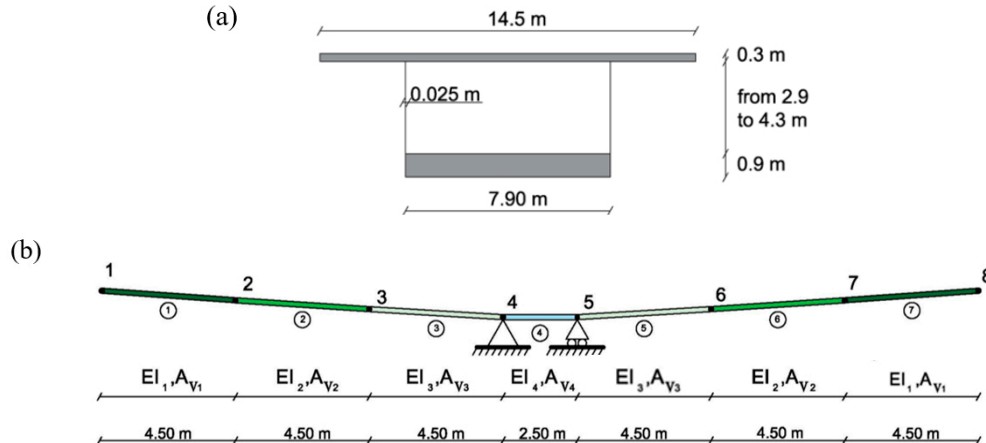

**Figure 11.** Yunbao Bridge: (**a**) cross-section and (**b**) finite element model (FEM).

The load of the formwork traveler considered in the structure is presented in Figure 12a. In this figure, the weight of the formwork traveler F (1041 kN) was assumed to be 60% of the maximum weight of the deck segment. The effect of each form traveler in the deck was simulated as a pair of vertical forces of values 0.25 F and 1.25 F (226 kN upward and 1267 kN downward). The analyzed loading case resulted from the combination of loads of the formwork traveler and the self-weight of the deck segments. The resultant loading case is presented in Figure 12b.

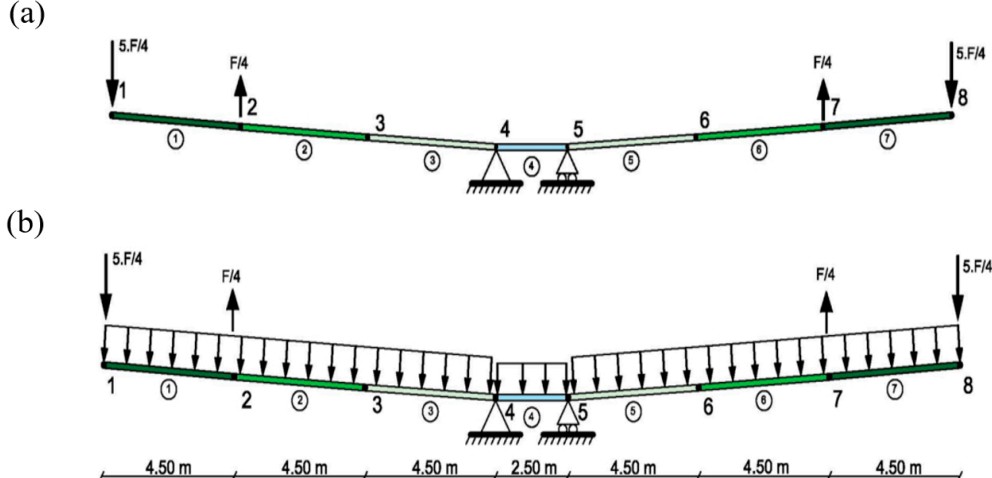

**Figure 12.** Loading case: (**a**) loads from the formwork traveler and (**b**) modeled loading case.

In Section 4, the calculated results of the effects of the shear rotations in the structure are discussed.

## 4. Discussion

The results of the direct analysis of the bridge are summarized in Figure 13. This figure compares the values of Midas/Civil with those obtained using the proposed methodology when shear deformation effects were considered in terms of deflections (Figure 13a) and rotations (Figure 13c). The latter included the rotations numerically calculated by the derivation of the vertical deflections, including shear effects throughout the beam directrix (axis *x*), i.e., $d(v_b + v_s)/dx$. The results of the shear diagram are also presented in Figure 13b. As the structure is isostatic, these diagrams were not affected by the effects of the shear deformations.

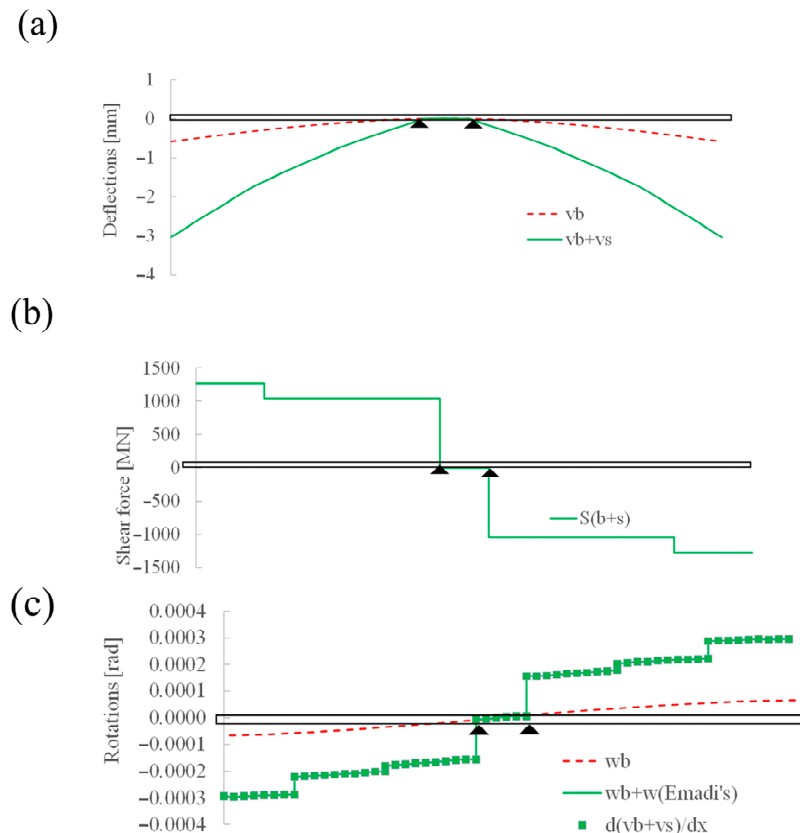

**Figure 13.** Yunbao bridge analysis: (**a**) vertical deflections, (**b**) shear diagram, and (**c**) rotations with and without shear effects.

The analysis of Figure 13a shows that the shear effects played an important role in the deformation of the structure. In fact, the deformation at the edge of the cantilevers increased from −0.6 mm to −3.1 mm when these effects were considered. Figure 13c shows that the rotations obtained using the numerical derivation corresponded exactly with those obtained using the proposed method, where the shear rotations ($w$(Emadi's)) were added to the bending rotations ($w_b$) calculated using the computer software. This figure also shows the effect of the shear diagrams in Figure 13b in the total rotations throughout the bridge deck.

A comparison of the magnitude of the obtained bending ($w_b$) and the shear rotations ($w_s$) on the left-hand side of the node are summarized in Table 4. This table also includes the percentage of shear rotations in terms of the total rotations ($w_b + w_s$) in the different nodes in the cantilever.

**Table 4.** Comparison of the rotations in the FEM.

| Rotation | $w_b$ (rad) | $w_s$ (rad) | $w_s/(w_b + w_s)$ |
|---|---|---|---|
| $w1,1$ | $-6.6 \times 10^{-5}$ | $-2.3 \times 10^{-4}$ | 77.7% |
| $w2,2$ | $-5.6 \times 10^{-5}$ | $-1.7 \times 10^{-4}$ | 74.8% |
| $w3,3$ | $-3.5 \times 10^{-5}$ | $-1.7 \times 10^{-4}$ | 82.7% |
| $w4,4$ | $-0.8 \times 10^{-5}$ | $-1.5 \times 10^{-4}$ | 94.5% |

The results in Table 4 show that the shear rotations should not be neglected in the analyzed deck segments. Obviously, the errors in rotations were related to the shear diagram. In fact, the higher the shear forces, the higher the shear rotations. For example, at the edge of the cantilever (nodes 1 and 7), the errors of neglecting the shear rotations could be as high as 77.7%. As the precamber given to the next segment is directly linked

with the value of this rotation, a proper calculation can help to provide a proper geometry, especially with segments that have such deformable webs under shear.

## 5. Conclusions

Traditional structural analysis software is based on Thimhosenko's stiffness matrix and, therefore, only considers shear effects on the vertical deformations of beams. In fact, the same rotations obtained by these programs are independent of whether this deformation is considered or not. This simplification might lead to significant errors in the simulation of the rotations in some structures (such as deep beams), as well as the wrong identification of the mechanical properties of the structural elements from the measurements on site (inverse analysis). Despite its important role, no other work studying the calculation of shear rotations in deep beams was found in the literature. To fill this gap, this study proposes a simple and intuitive method to calculate the shear rotations in both isostatic and statically redundant beams for first-order shear deformation theory. No examples of frames composed of deep beams are presented here. However, this methodology can also be applied to such structures.

A set of parametric analyses were carried out to determine the slenderness ratios from which the shear rotations could be neglected in different structural systems. In addition, the errors in the inverse analysis of deep beams were parametrically studied. The importance of the proposed methodology was highlighted in the simulation of a construction stage of a composite bridge. The obtained results show the important role that shear rotations play in deep beams. The effect of prestressing on shear rotation will be addressed in future studies.

**Author Contributions:** Conceptualization, methodology, software, writing—original draft preparation, formal analysis and validation, S.E.; investigation and writing—review and editing, J.A.L.-G.; resources and writing—review and editing, H.M.; supervision and project administration, J.T. All authors have read and agreed to the published version of the manuscript.

**Funding:** The authors are indebted to the projects PID2021-126405OB-C31 and PID2021-126405OB-C32 funded by FEDER funds—A Way to Make Europe and Spanish Ministry of Economy and Competitiveness MICIN/AEI/10.13039/501100011033/, BIA2017-86811-C2-1-R and BIA2017-86811-C2-2-R funded by FEDER funds.

**Institutional Review Board Statement:** Not applicable.

**Data Availability Statement:** The data presented in this study are available on reasonable request from the corresponding author.

**Conflicts of Interest:** The authors declare no conflict of interest. The funders had no role in the design of the study; in the collection, analyses, or interpretation of data; in the writing of the manuscript; or in the decision to publish the results.

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
