# Peer review of "Simplified Calculation of Shear Rotations for First-Order Shear Deformation Theory in Deep Bridge Beams"

_applsci, doi:10.3390/app13053362_

Round 1
Reviewer 1 Report
The manuscript presents an interesting issue dealing with shear rotations in deep bridge beams, and the paper is well written and meet the scope of the journal. The detailed comments are as follows:
1. The abstract should emphasize the innovation of the research, but the authors overemphasize the necessity of the research, please revise it.
2. Please explain the difference between your calculation method and existing methods.
3. The proposed method can calculate the elastic shear rotation; can it be extended to calculate the plastic shear rotation of deep beams? Please clarify the engineering application prospect of the proposed method.
Author Response
A document answering the Reviewer's comment is uploaded.

Reviewer 2 Report
The reviewer thanks the authors for their contribution. The contents of the article are interesting and useful information for structural engineering applications. Nevertheless, the publication in the “Applied Sciences, MDPI” is not recommended unless the following suggestions are taken into account:
1) Shear rotations in three point bending tests of slender beams spring out by increasing the vertical concentrated load, i.e., the Euler-Bernoulli model starts to deviate in favor of that of Timoshenko. Please, consider this point within the article. Analytical simulations of slender beams under the increment of the vertical load would be appreciated.
2) Based on the following reference, the shear effects are also not negligible along beams with higher values of slenderness. Please analyze and refer to the corresponding contents. Additional simulations would be appreciated:
- https://doi.org/10.1061/(ASCE)EM.1943-7889.0001643
3) The authors refer to “Deep Bridge Beams” but, conversely, the effect of prestressing is never taken into account. In short, the authors consider the first-order beam theories of the Euler-Bernoulli and Timoshenko model. Title and contents of the article should be revised based on this issue.
4) Based on the following reference, the second-order beam theory of the Euler-Bernoulli model is valid for slender prestressed concrete girder-bridges. The static behavior of “Deep Concrete Bridge Beams”, i.e., under the effect of prestressing, could be analyzed in further investigations:
- https://doi.org/10.12989/sem.2021.77.1.001
5) According to the comments reported at point 3, it is not appropriate referring to “balanced cantilever bridges” in Section 3.3 which, in fact, are superstructure that are sequentially joined by segments to form spans using post-tensioning (i.e., there is the effect of prestressing). Please revise the corresponding parts within the article.
Author Response

(The authors gave the same response as above.)

Round 2
Reviewer 1 Report
This manuscript is recommended for acception in the present form.Reviewer 2 Report
The authors carried out the required revisions.